# Cross-Protection of Hepatitis B Vaccination among Different Genotypes

**DOI:** 10.3390/vaccines8030456

**Published:** 2020-08-16

**Authors:** Takako Inoue, Yasuhito Tanaka

**Affiliations:** 1Department of Clinical Laboratory Medicine, Nagoya City University Hospital, Nagoya 467-8602, Japan; clinoue@med.nagoya-cu.ac.jp; 2Department of Virology and Liver Unit, Nagoya City University Graduate School of Medical Sciences, Nagoya 467-8601, Japan; 3Department of Gastroenterology and Hepatology, Faculty of Life Sciences, Kumamoto University, Kumamoto 860-8556, Japan

**Keywords:** hepatitis B virus, genotypes, universal vaccination, vaccine escape mutants

## Abstract

Hepatitis B (HB) vaccination is the most effective method for preventing HB virus (HBV) infection. Universal HB vaccination containing recombinant HB surface antigens (HBsAg) is recommended. Our data revealed that human monoclonal HB surface antibody (anti-HBs) from individuals inoculated with genotype C-based HB vaccine induced cross-protection against HBV genotype A infection. An in vitro infection model demonstrated anti-HBs-positive sera from individuals inoculated with genotype A- or C-based HB vaccine harbored polyclonal anti-HBs that could bind to non-vaccinated genotype HBV. However, because there were low titers of anti-HBs specific for HBsAg of non-vaccinated genotype, high anti-HBs titers would be required to prevent non-vaccinated genotype HBV infection. Clinically, the 2015 Centers for Disease Control and Prevention guidelines state that periodic monitoring of anti-HBs levels after routine HB vaccination is not needed and that booster doses of HB vaccine are not recommended. However, the American Red Cross suggests that HB-vaccine-induced immune memory might be limited; although HB vaccination can prevent clinical liver injury (hepatitis), subclinical HBV infections of non-vaccinated genotypes resulting in detectable HB core antibody could not be completely prevented. Therefore, monitoring anti-HBs levels after routine vaccination might be necessary for certain subjects in high-risk groups.

## 1. Introduction

It is well known that the hepatitis B virus (HBV) is one of the most common blood-borne viral infections worldwide. Chronic hepatitis B (CHB) caused by HBV infection affects approximately 260 million persons worldwide [1]. Although current antiviral treatments using nucleos(t)ide analogs or pegylated interferons are effective for CHB, HBV cannot easily be eliminated from the liver, because its relaxed circular DNA is persistently converted to covalently closed circular DNA (cccDNA) in the cell nucleus [2]. Most CHB patients have favorable clinical outcomes, but approximately 15–40% of patients develop cirrhosis and/or hepatocellular carcinoma (HCC) [3]. Therefore, the control of HBV infection is extremely important. Approaches to prevent HBV infection include disturbing the route of transmission and treating chronically-infected patients as well as susceptible individuals with immunoprophylactic treatments [4].

In 1982, two HB vaccines were licensed from France and the United States [5]. Initially, the HB vaccine prevented HBV infection efficiently [5,6] and by the end of 2015, 185 countries had included hepatitis B vaccination in their national Expanded Program on Immunization [7]. Current HB vaccines are highly effective with a protective rate of 94–98% against chronic HBV infection for at least 20 years [8].

HBV strains are classified into 10 genotypes based on genetic diversity [9] and the prevalence of these genotypes varies geographically [10]. The first four genotypes of HBV (A, B, C, and D) were proposed and divergence above 8% of the complete genome was designated as a criterion for genotype identification [10,11]. Almost simultaneously, genotyping based on phylogenetic clustering was recommended [12]. By sequencing the HBsAg coding region, four new strains were designated as novel genotypes E and F. This added a new criterion for genotyping: 4% of nucleotide divergence in the HBsAg coding sequence [13]. Genotype F was defined by the analysis of the full genome sequence [14]. Subsequently, genotypes G [15], H [16], I [17], and J [18] were defined. Hepatitis B surface antigen (HBsAg) is the fundamental molecule for HBV entry into hepatocytes [19] and HBV vaccination establishes host immunity by activating B lymphocytes that produce neutralizing HBsAg-specific antibodies (anti-HBs). The highly immunogenic region of HBsAg, known as the “a” determinant, contains two peptide loops in which several amino acids vary among the HBV genotypes [20].

In this review, we provide information regarding the cross-protection of hepatitis B vaccination among different genotypes. In addition, we discuss the features of each HBV genotype, prevention against non-vaccinated genotype HBV infection, and the requirement of booster HB vaccination.

## 2. HBV Serotypes and Genotypes

### 2.1. Geographical Characteristics of HBV Genotype Distribution

Based on some of the antigenic determinants of HBsAg, there are nine serological types referred to as subtypes *adw2, adw4, adrq+, adrq-, ayw1, ayw2, ayw3, ayw4*, and *ayr*. [21,22].

Ten HBV genotypes (A–J) have been recognized. They characterize a particular geographical distribution [9,21,23]. The features and geographical distributions are shown in Table 1. Genotype A is often found in India, Africa, North America, and northwestern Europe [24]. There are many patients infected with HBV genotypes B and C in Asia. In Eastern Europe and the Mediterranean, HBV genotype D is important [10]. Genotype E is predominant in central and western Africa [25]. Genotype F is major in South America and Mexico [26]. Genotype G is often found in the United States, France, Germany, and Central America. Genotype H dominates in Central America [27]. Genotype I is found in Vietnam. Genotype J, which is a possible recombination with genotype C, was reported from Japan [18].

### 2.2. Clinical Features Related to Differences in Genotype

The prevalence of HBV genotype A is significantly higher in the men who have sex with men (MSM) population compared with the rest of the population [28]. HBV subgenotype A2 and genotype C are likely to prevail in populations at high risk of infection via sexual transmission [29,30]. In addition, HBV genotype A advances into a persistent infection more often than genotype C [31,32]. Infection with HBV/Ba, where recombination with genotype C occurs in the basal core promoter (BCP) region [33], has been found in many Asian countries, such as Taiwan, and is associated with a higher risk of developing HCC in HBV carriers [34].

Genotypes C and F are associated with disease severity and response to treatment [35,36,37]. Compared to subgenotypes B2 and C2, subgenotype C1 shows better antiviral response to nucleos(t)ide analogs in patients positive for hepatitis B envelope antigen (HBeAg) [38]. Patients infected with genotypes C and F have a higher prevalence of HCC than those infected with genotypes B and D [39]. Subgenotypes in HBV genotype C may be associated with the increase of HCC occurrence in patients with HBeAg-positivity [40]. Patients infected with HBV genotype C progress rapidly to cirrhosis and HCC than patients infected with genotype B in Asia [41,42,43,44]. Moreover, more patients infected with HBV genotype D developed to cirrhosis and HCC than patients infected with HBV genotype A [24,45,46,47].

**Table 1 vaccines-08-00456-t001:** The features and geographical distribution of each genotype HBV.

Genotype	Serotypes [48]	Geographical Distribution (Subgenotype)	Major Clinical Specificity	Major Biological Finding	References
A	adw	Asia, Africa (A1)	Sexual transmission (MSM), Chronic infection	BCP double mutation 1762T/1764A, 1888A, 1802–1803CG, 1858C	[28,31,32,49,50]
	adw	Northwest Europe, North America (A2)	Sexual transmission (MSM)	BCP double mutation 1762T/1764A, 1802–1803CG	[29,30,50]
B	adw, ayw	Tohoku, Hokkaido, Okinawa in Japan (Bj)	High rate of spontaneous HBeAg seroconversion	Precore stop codon mutations	[49,50,51]
	adw, ayw	Asia (Ba)	Higher risk of developing HCC	Recombination with genotype C, 1802–1803TT, 1858T	[33,49,50]
C	ayw, ayr, adr	Asia, Japan	Mother-to-infant transmission, Rapid progression to cirrhosis and HCC	1802–1803TT, 1858C (genotype C1), 1858T (genotype C2)	[29,30,39,41,42,43,44,50]
D	ayw	Western Europe, Mediterranean	Histological inflammation, Early HBeAg seroconversion	BCP double mutation 1762T/1764A, 1802–1803CG, 1858T, T1764G1766 core promoter double mutants	[36,50,52]
E	ayw	West Africa	High viral loads, high frequency of HBeAg-positivity	1802–1803CG, 1858T	[25,50]
F	adw	Alaska, Mexico, South America	HCC occurrence	1802–1803TT, 1858C	[39]
G	adw	France, Germany, USA	Chronic infection, MSM	A 36-nucleotide insert, 3′ of position 1905, two translational stop codons at positions 2 and 28 of the precore/core region	[50]
H	adw	Central America	HCV coinfection and obesity are common cofactors.	1858C	[26,27,50]
I	adw	Vietnam	(Few clinical features)	A recombinant of genotypes A/C/G	[17]
J		Japan	Isolated from a single Japanese man with HCC	A recombinant of genotype C and gibbon HBV in the S region	[18]

Abbreviations: HBV, hepatitis B virus; MSM, men who have sex with men; BCP, basal core promoter; HBeAg, hepatitis B envelope antigen; HCC, hepatocellular carcinoma; USA, United States of America.

### 2.3. Major Biological Features Related to Differences in Genotype

Independent from the HBeAg status, 1762T/1764A was significantly more frequent in genotypes A and C compared with D and B [24,49]. The BCP 1762T/1764A mutations are not selected in the subgenotype Bj/B1 and B6 strains because of the 1896A and precore stop codon mutations [51]. The BCP double mutation 1762T/1764A occurs in approximately 25% of cases [53] without differences between subgenotypes A1, A2, and genotype D. The mutation G1896A, which leads to a stop codon that truncates HBeAg during translation, does not develop in the genotypes/subgenotypes in which 1858C occurs frequently [54,55] and with which 1858C is positively associated, namely, A, C2, F2, and H [50]. G1896A is frequent in genotypes/subgenotypes with 1858T, C1, D, E, and F [50,56]. More than 40% of genotype D cases had G1896A [57]. Overall, the frequency of G1896A is 100% in genotype G, 50% in genotype B, 40% in genotype D, and 23% in genotype C [56]. Thus, the estimated annual rate of HBeAg to anti-HBeAg seroconversion can differ between genotypes [58,59].

1762T/1764A and 1896A were more frequent in genotype D isolates from HBeAg-negative individuals compared with HBeAg-positive individuals [24]. Even though the precore region of genotypes D and E cannot be differentiated and both have 1858T, they do not develop G1896A at the same frequency. The high frequency of HBeAg-negativity in genotype D is a result of G1896A [60,61], which has statistically positive and negative correlations with genotypes D and E, respectively [50]. This was confirmed by a comparison of the frequencies of G1896A [53], where G1896A occurred in 47.2% of genotype D sequences compared with 34.2% of genotype E sequences (*p* < 0.0001). T1764G1766 core promoter double mutants are restricted to hepatitis B virus strains with A1757 and are common in genotype D [52].

Genotype G is unique in that all sequences have a premature stop codon at position 2 of the precore precursor protein and therefore HBeAg is not expressed [15]. Genotype I is a recombinant of genotypes A/C/G, which clusters close to genotype C when the complete genome is analyzed and with genotype A when the polymerase is analyzed [62].

## 3. Transmission and Protection

### 3.1. Transmission

HBV infection occurs through various forms of human contact. In addition to vertical transmission from mother to newborn, close domestic contact, needle sharing, sexual contact, and occupational (healthcare) exposure are included as horizontal transmission [63]. HBV is transmitted generally via the permucosal or percutaneous contact to body fluids containing HBV. The major risk factor related to HBV sexual transmission is unprotected sex with a partner infected with HBV (heterosexual or homosexual). In this situation, 26.1% of sexual partners showed evidence of past and/or current HBV infection [64]. When a non-HBV immune individual is exposed to blood from a patient positive for HBeAg or someone who has HBV DNA >10^6^ IU/mL, the risk of HBV transmission is estimated at 19–30% [65]. The blood (serum) is the most serious source of HBV infection [66]. HBV transmission can be occurred by the unintentional inoculation of small volumes of blood or other body fluids during medical techniques [39]. Now, blood transfusion and organ transplantation are extremely uncommon ways for HBV transmission.

HBV is efficiently infected by sexual transmission [63]. The main risk factors are unprotected sex with a partner infected with HBV, including unvaccinated MSM, heterosexual individuals with multiple sexual partners, and sex workers [39].

However, as shown by the 40% transmission rate to nonimmune partners of patients with acute hepatitis B or chronic HBV infection, heterosexual transmission is still important [67,68]. The seroprevalence rates of HBV-related biomarkers are positively associated with the number of current and lifetime heterosexual partners [69,70].

### 3.2. Protection

For the prevention of HBV infection, both HB vaccines and hepatitis B immunoglobulin (HBIG) have been approved [71,72]. HBIG which is prepared from human plasma containing a high concentration of anti-HBs (e.g., 20,000 IU per dose [73]) makes available short-period (3–6 months) protection from HBV infection. Generally, HBIG is used for post-exposure prophylaxis along with HB vaccination for individuals who have never been vaccinated or who have not responded to HB immunization. The suggested dosage of HBIG is 0.06 mL/kg [74].

The features and effects of HB vaccination are shown below. For pre-exposure prophylaxis, HB vaccination is suggested for all unvaccinated children and teenagers, all unvaccinated adults at high risk of HBV infection (particularly MSM, adults with various sexual partners, and drug users), and all adults who desire protection from HBV infection [72]. In 1992, the World Health Organization (WHO) recommended that all countries should introduce universal HB vaccination into their routine immunization programs [75]. Vaccination has demonstrated highly successful in reducing the disease burden, the development of carrier states, and hepatitis B-related morbidity and mortality [76].

Unimmune individuals or those known not to have received a full HB vaccine series should receive both HBIG and HB vaccines as soon as we can (preferably ≤24 h) after exposure to blood or body fluids which contain HBsAg [77]. HB vaccine should be administered at the same time as HBIG but at an isolated injection site. The HB vaccine series should be completed, using the age-appropriate vaccine dosage and program [77]. Serologic testing of anti-HBs, HBsAg, and alanine transaminase (ALT) levels should be tested immediately after exposure and rechecked within 3–6 months. Both passive-active postexposure prophylaxis (simultaneous administration of HBIG and HB vaccine at separate sites) and active postexposure prophylaxis (administration of HB vaccination alone) are highly effective at preventing HBV infection [71]. Individuals with certification that they received a complete HB vaccine series and who have never experienced post-vaccination serologic testing should receive a single vaccine booster dose. These individuals should be treated in relation to the guidelines for the management of individuals with occupational exposure to blood or body fluids that contain HBV [78].

## 4. The Features and Effects of HB Vaccination

### 4.1. The Features of HB Vaccines

HB vaccines contain HBsAg that are produced by a recombinant yeast strain [79]. Epidemiologic studies have not found any evidence of an association between HB vaccination and sudden infant death syndrome or other causes of death during the first year of life [80,81]. Therefore, HB vaccination can be considered safe.

HB vaccination is available for younger children, adolescents, and healthy adults [77] and is the most effective way of preventing HBV infection [5]. The introduction of universal HB vaccination for newborns was reported to be a very reasonable and cost-effective strategy [82,83]. The WHO has now included HB vaccination in the Expanded Program on Immunization [84] and recommends that all infants receive a HB vaccine as soon as possible after birth, preferably within 24 h. In 2017, more than 180 WHO member states immunized infants against HBV as part of their routine vaccination schedule, and 84% of children received HB vaccines [1]. Because available data show that current HBV-A2 vaccines are highly effective at preventing infections and clinical disease caused by all known HBV genotypes, recombinant HB vaccines containing HBsAg generated from HBV genotype A2 have been used worldwide [85]. However, the remaining problem is vaccination lag. Africa is the least-vaccinated area against hepatitis B. Only one in ten infants in Africa are vaccinated at birth, suggesting that new vertical infection is still present in African countries [86].

### 4.2. The Effects of HB Vaccination

According to the data in China, compared with the eastern (6.6%) and central (5.2%) regions, the prevalence of HBV infection was the highest in western China (8.9%), which is considered a high prevalence area [87]. This conclusion is consistent with an epidemiological survey in 2007 [88]. Based on a report released in 2012, the immunization coverage of suitable birth dose varied widely from 94% in Beijing to only 25% in Tibet, and the three doses of HB vaccine varied from 100% in Beijing to only 79% in Tibet [87]. These results show that plans including a timely birth dose of HB vaccine and immunization coverage of three doses of HB vaccine should be strictly followed.

It is widely recognized that an anti-HB surface antibody (HBs) level ≥10 mIU/mL is protective against HBV infection [63]. HB vaccine-induced immune memory can be kept last for more than 20 years [89,90,91]. In youths and healthy adults (aged younger than 40 years), about 30–55% of recipients achieve protective antibody responses (i.e., anti-HBs ≥10 mIU/mL) after the first vaccination, 75% after the second vaccination, and over 90% after the third vaccination. Consequently, HB vaccination induces protective antibody responses (anti-HBs ≥10 mIU/mL) in most recipients.

Approximately 5–10% of healthy individuals, a series of HB vaccination fails to produce protective anti-HBs levels (>10 mIU/mL) [92]. Increasing age, smoking status, male gender, and obesity are the risk factors for poor or no response to HB vaccines [92]. Specific human leukocyte antigen (HLA) types have been reported to be associated with anti-HBs response to HB vaccination [93]. The HLAs are coded by the major histocompatibility complex (MHC) group of genes located on chromosome six in the human genome. In efforts to overcome the low and non-responsiveness to HB vaccination, several approaches have been proposed. An additional dose, an additional three-dose series, an increased vaccine dose, changing the route of administration, new adjuvants, and granulocyte-macrophage colony-stimulating factor (GM-CSF) have all been proposed to be prominent factors for improving the seroprotection rate [94]. Komatsu described that the most common strategy for low responders and non-responders is to give an additional vaccine or a series of vaccines [94]. The best injection site was confirmed as the deltoid muscle, except for infants [95]. Regarding low- and non-responders to the initial three-dose series, 39–91% and 61–100% showed good responses after one additional dose (4th dose) or an additional three-dose series, respectively [96,97]. An additional three high-dose series vaccine could further improve the seroprotection rate [98].

Regardless of specific patient considerations when an HB vaccination schedule is selected, a complete vaccine series should be administered [74]. Recommendations on the HB vaccine dosage and schedule differ depending on the product used and the recipient’s age [77]. Details on HB vaccination are described in guidelines “Guidelines for the Prevention, Care and Treatment of Persons with Chronic Hepatitis B Infection” and “Recommendations of the Advisory Committee on Immunization Practices (ACIP) Part II: immunization of adults” [39,72]. The requirement for booster vaccination is discussed in a later section (The requirement for booster vaccination).

## 5. Induction of Cross-Genotype Protection by HB Vaccination

Genotype A HBV is dominant in North America, northwestern Europe, India, and Africa and HBV genotype A2-type recombinant HB vaccines are effective at preventing non-genotype A2 HBV infections [85]. In contrast, genotype B and genotype C HBV strains are predominant in East Asian countries [99]. Some of these countries, including Japan and Korea, have used recombinant HB vaccines produced from genotype C for immunoprophylaxis against specific HBVs prevalent in their communities [100]. In Japan, the spread of genotype A HBV strains introduced from foreign countries and the subsequent increase of hepatitis caused by genotype A HBV is of increasing concern [101].

Since the 1980s, the effect of cross-immunity by HB vaccines of different genotypes has been studied. Epidemiological studies report that vertical transmission and/or incident infection with non-genotype A2 HBV was efficiently prevented in countries with universal childhood vaccination programs using the genotype A2 HBV vaccine [85]. Furthermore, a single mouse monoclonal antibody protected chimpanzees from infection by both *adr* (genotype C HBV) and *ayw* (genotype D HBV) strains [102]. Next, we present our data on the neutralizing capacity of HB vaccines against different genotypes of HBV.

### 5.1. Prevention of Genotype A Strain Infection and Vaccine Escape Mutant Infection by Genotype C-Derived HB Vaccines In Vitro and In Vivo

Previously, we isolated human monoclonal antibodies (mAbs) against HBV from healthy volunteers who had been immunized with a genotype C type recombinant HBV vaccine (Biimugen), using a cell-microarray system [103,104,105]. Among these mAbs, HB0116 and HB0478, which recognized the first N-terminal peptide loop within the “a” determinant, had HBV-neutralizing activity [106]. We also showed that HB0116 and HB0478, produced by immunization with the genotype C type vaccine, neutralized the infectivity of genotype C and genotype A HBV [107]. In vitro experiments showed that HB0478 at doses above 55 mIU completely protected against genotype C and genotype A infection, whereas HB0478 at a lower dose of 5.5 mIU protected against genotype C infection only (Figure 1).

The analysis of nine HBV DNA positive blood donors in the United States revealed that 5 individuals immunized with genotype A2-type vaccine were not protected against infections by non-genotype A2 HBV [108]. However, the serum anti-HBs levels of these individuals (3–96 mIU/mL) were relatively low. Remarkably, the infections remained at a subclinical level in these vaccinated individuals, who subsequently resolved the HBV infection, suggesting that genotype A2 vaccination did not prevent non-genotype A2 infection but did inhibit the development of clinical manifestations [108]. Thus, HBV-specific antibodies induced by genotype C vaccines might protect against clinical hepatitis caused by infection with non-genotype C genotypes, even with lower anti-HBs concentrations. In conclusion, our study demonstrated that active immunization with a genotype C-based vaccine might confer prophylaxis against genotype A and against escape mutants such as G145R (as described later) when the anti-HBs responses are sufficient [107].

### 5.2. Prevention of Genotype C HBV Infection with a Genotype A-Derived Vaccine, and Genotype A HBV Infection with a Genotype C-Derived Vaccine In Vitro

In an in vitro infection model, anti-HBs-positive sera from individuals inoculated with genotype A- or C-based HB vaccines harbored polyclonal anti-HBs that bound to non-vaccinated genotype HBV. However, because titers of anti-HBs specific to the HBsAg of the non-vaccinated genotype were low (<30 mIU/mL), high anti-HBs titers (≥50 mIU/mL) would be required to prevent non-vaccinated genotype HBV infection [109].

To examine whether vaccine-acquired polyclonal anti-HBs effectively bind to the HBsAg of non-vaccine genotypes, we established genotype A-antigen and genotype C-antigen ELISAs, which were considered accurate for measuring the reactivity to single genotype-derived HBsAgs [109]. The results showed that vaccine-acquired polyclonal anti-HBs with high titers could prevent infection by vaccine and nonvaccine genotypes HBV equally [109]. This was consistent with previous epidemiological studies showing that single genotype-derived HBsAg vaccination effectively decreased multiple genotype HBV infections [84,110,111,112]. This result is also in accordance with our previous in vitro study, which showed that monoclonal anti-HBs prevented HBV infection of non-vaccine genotype at high titers [107]. These findings indicate that neutralizing capacity related to common major epitopes such as the ‘‘a’’ determinant and C(K/R)TC motif (121–124 amino acids [aa]) is critical for preventing HBV infection of multiple genotypes [113,114].

Taken together, these findings indicate that high titers of anti-HBs are required to prevent infection with nonvaccine genotype HBV [109]. In conclusion, single genotype HBsAg vaccination with genotype A-derived and genotype C-derived vaccines induce polyclonal anti-HBs that bind appropriately to non-vaccine HBsAg. The vaccine-acquired polyclonal anti-HBs sufficiently neutralized non-vaccine genotype HBV at a high anti-HBs titer; however, the neutralization was unsatisfactory at a low anti-HBs titer. Therefore, it is recommended to maintain high anti-HBs titers to prevent infection by non-vaccine genotype HBV as well as vaccine genotype HBV [109].

## 6. Vaccine Escape Mutants

### 6.1. The HBsAg “a” Determinant

All HBV genotypes and serotypes share the common “a” determinant, which spans 124–149 aa within the major hydrophilic region and which forms two major loops and one minor loop (A-loop) with cysteine-disulfide bonds that extend from the outer surface of the virus. A second hydrophilic loop (139 to 147 or 149 aa) is the major target for neutralizing (protective) anti-HBs produced following natural infection or vaccination [113] and this provides protection against all HBV genotypes and subtypes and is responsible for the broad immunity afforded by HBV vaccination (Figure 2). Thus, alterations of residues within this region of the surface antigen can induce conformational changes that allow the replication of mutated HBVs in vaccinated individuals (vaccine escape mutants). In addition, such mutated HBVs can be undetectable by existing diagnostic assays, posing a potential threat to the safety of blood supplies [113,115].

HBV strains with amino acid substitutions in HBsAg often escape from HB vaccine-induced antibody and HBIG treatment during the vertical transmission of HBV [116,117,118]. The most frequently reported substitution is residue 145 (glycine to arginine, G145R), located in the second loop of the “a” determinant of HBsAg.

### 6.2. S-Gene Mutants

Representative S-gene mutants are shown in Table 2. An important mutation in the surface antigen region was first recognized in Italy 30 years ago [119,120]. This virus has a G587A mutation, resulting in a G145R substitution of the “a” determinant of the surface antigen. Because the G145R substitution changes the projecting second loop of the “a” determinant, neutralizing antibodies induced by vaccination no longer recognize the mutated epitope [121]. Under the immune pressure of HB immunization, especially when HBIG is combined, HBV with mutations in the “a” determinant can be selected [120].

The major vaccine escape mutants reported in Taiwan [122,123,124,125,126,127], Japan [128,129,130,131,132], Singapore [133], England, Wales [134], China [135,136], Pacific Islands [137] and Indonesia [138] are shown in Table 2. In addition to the prototype G145R, other S-gene mutations (alone or in combination) that might escape neutralizing anti-HBs have been identified worldwide (Table 2) [123,125]. HBV infection with S-gene mutation has been reported in the presence of protective levels of anti-HBs in infants born from HBV-infected mothers who received HBIG plus HB vaccine, in liver transplant patients who received HBIG for prophylaxis, and in HBsAg-negative HBV carriers. Globally, the emergence of G145R is a rare event more generally associated with the use of HBIG rather than HB vaccination [124,139].

In 2010, a survey conducted in Taiwan reported vaccine escape mutants were not increased in a population of children and adolescents who were fully covered by universal infant immunization over 20 years [124]. However, a recent study in China reported that HBV mutants capable of infecting people emerged 13 years after the implementation of a successful universal vaccination program [140]. However, a more careful analysis of the data from that study showed that following vaccination, both the HBsAg carrier rate and prevalence of variants indeed decreased, even though the variant prevalence decreased at a lower rate (71% versus 33%) [141,142].

In Taiwan, the baseline prevalence of a mutant was 7.8% (8/103) in HBsAg carrier children, and was maintained at around 20% (19.6% [10/51] to 28.1% [9/32]) among HBsAg carrier children in the first 15 years of the universal mass vaccination program [120]. In the last 10 years, there has been no steady increase in vaccine escape HBV mutants in Taiwanese carrier children who failed in the mass vaccination program, and there is no evidence of the spread of this virus, likely because of the weakness of the mutant virus [143]. Despite the increased percentage of surface gene mutants after mass hepatitis B vaccination, the actual number of children infected with this mutant is small and is not increasing [120]. Studies in Italy also reported the same conclusions [144]. Therefore, the presence of vaccine escape mutants does not seem to threaten the ongoing hepatitis B control strategies in Taiwan and Italy, and perhaps, worldwide. Therefore, the currently available hepatitis B vaccines can be continued.

## 7. The Requirement for Booster Vaccination

HB vaccine-induced anti-HBs declines rapidly in the first year and then more slowly [5]. Over time, the anti-HBs often becomes undetectable. However, vaccine-induced immunologic memory is conserved for at least 12–18 years despite the decline of anti-HBs [145,146,147].

According to the 2015 CDC guidelines, the periodic detecting of anti-HBs levels after routine HB vaccination is not necessary and booster doses of HB vaccine are not currently recommended [77]. Actually, a booster vaccination is not needed for at least 20 years in Taiwan: surveillance did not reveal any increase in acute hepatitis B [148] or chronic HBV infection [8] in adolescents vaccinated 20 years ago. For prevalent areas where the primary goal of HB immunization is to prevent HBV infection in infancy [83], even if immunity conferred by the vaccine given in early childhood disappears, when unprotected vaccinated individuals contract HBV infection in adulthood, the risk of becoming an HBsAg carrier is low [149].

On the other hand, the American Red Cross reported that immune memory induced by HB vaccines might be deficient. Although HB vaccination prevented the occurrence of clinical liver injury (hepatitis), subclinical infections cannot be prevented perfectly [108]. Stramer et al. reported that most individuals with low anti-HBs titers (5 out of 6 cases) and who were HBV-DNA positive 7–16 years after vaccination, were exclusively or generally infected with non-vaccine genotype HBV [108]. These cases may be examples of vaccine recipients who were infected with non-vaccine genotype HBV due to the time-dependent reduction in polyclonal anti-HBs. Our in vitro, in vivo, and alignment data among HBV genotypes (Figure 2) might show possibly escape from HBV vaccination in individuals with low anti-HBs titers.

Therefore, to maintain anti-HBs titers, booster vaccination has been recommended, particularly for cohorts with a high risk of HBV infection such as immunocompromised patients and health care providers [150,151,152,153]. When the serum anti-HBs level is not enough to prevent HBV infection (anti-HBs <10 mIU/mL), a booster vaccination should be performed. Because HB vaccines are highly immunogenic, postvaccination serologic testing is not necessary. However, the testing might be introduced for the following individuals; infants whose mothers were infected with HBV, individuals with occupational risk of exposure to blood (e.g., health care workers, policemen, and firemen), sexually active individuals such as MSM, or immunosuppressed patients (receiving hemodialysis, organ transplantation, and blood transfusion) [63,154].

To be sure, although the HB vaccine is appropriately effective at preventing the development of hepatitis (clinical disease), it cannot prevent 100% of HBV infections, leading to anti-HBc-positivity [108]. In addition, considering the induction of cross-genotype protection by HB vaccination described above, booster vaccination might be useful for preventing infection of non-vaccine genotype HBV in individuals with low anti-HBs titers.

## 8. Discussion

HB vaccination is clearly very successful at preventing and controlling hepatitis B and HBV-related diseases globally. Anti-HBs responses gradually decrease after a single course of vaccination. Immunocompromised hosts, such as hemodialysis patients and HIV-infected patients, are known low responders to HB vaccines [94]. However, there are no standard guidelines for the necessity, timing, or method of booster vaccination under various situations. Cases of hepatitis B in fully vaccinated individuals are rare. Breakthrough infections caused by S-gene mutants are occasionally reported but currently, they do not pose a serious threat to the established vaccination programs. However, the emergence of drug-resistant mutants with alterations in the “a” determinant of the S-protein is of concern.

Our data revealed that human monoclonal anti-HBs from individuals who inoculated genotype C-based HB vaccine can induce cross-protection against HBV genotype A infection [107]. Meanwhile, according to an in vitro infection model, anti-HBs-positive sera from individuals who inoculated genotype A or C-based HB vaccine have polyclonal anti-HBs that sufficiently bound to non-vaccinated genotype HBV [109]. However, because anti-HBs specific to the HBsAg with non-vaccinated genotype were relatively small amounts, enough anti-HBs titers would be required to prevent non-vaccinated genotype HBV infection.

Additionally, S-gene mutant HBV may escape from HB vaccination with S-protein alone. To prevent the infection and spread of HBV with an S-mutation, the effectiveness of third-generation HB vaccines containing pre-S-proteins in addition to S-protein should be determined. Several countries have not yet introduced universal vaccination programs. The promotion of universal vaccination in these countries is mandatory for the global eradication of HBV infection.

The significance of the co-administration of HBIG with HB vaccine to prevent the mother-to-infant transmission of HBV needs to be fully evaluated. Although the administration of HBIG prevented the intrauterine transmission of HBV and reduced overt infantile hepatitis [126,155], appropriate randomized control trials should be performed in the future.

Clinically, according to the 2015 CDC guidelines, the booster doses of HB vaccine are not recommended. Conversely, the American Red Cross and Xu et al. [156] suggests that HB-vaccine-induced immune memory might be restricted. Because transient infection can occur in individuals whose levels of anti-HBs do not exceed >10 mIU/mL, careful follow-up studies are needed to prevent individuals from acute clinical hepatitis and chronic infection. In our opinion, serologic testing after HB vaccination, especially anti-HBs, should be performed for individuals at high risk for HBV infection. The requirement for booster doses to reserve vaccine-induced immunity should be assessed regularly, particularly for infants whose mothers were with HBV infection, individuals with occupational risk (e.g., health care workers, policemen, and firemen), sexually active individuals, or individuals under immunosuppression.

## Figures and Tables

**Figure 1 vaccines-08-00456-f001:**
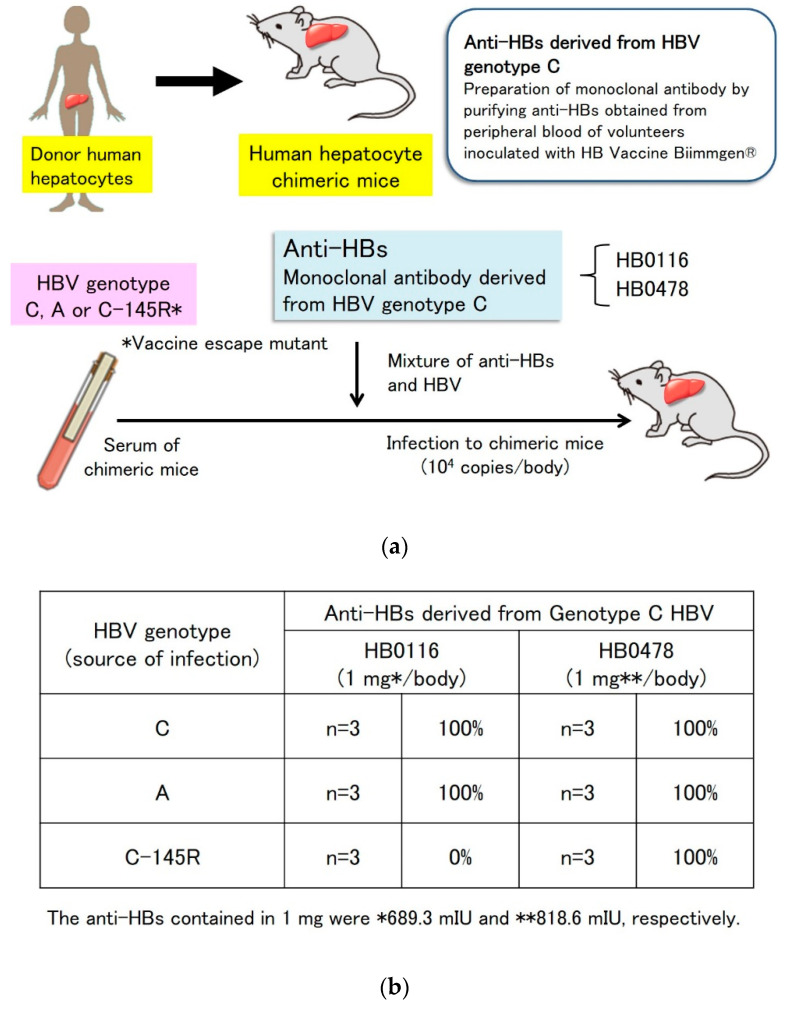
HBV infection protection test using chimeric mice. (**a**) HBV infection protection test using anti-HBs derived from genotype C HBV in chimeric mice. Chimeric mice were inoculated with these culture supernatants to obtain monoclonal and intact infectious virions. After establishing viremia in these mice, the sera were collected and used as inocula after titration in another experimental chimeric mouse. Five weeks after injection, serum HBV DNA was measured by quantitative PCR. (**b**) Protection rate against HBV infection in chimeric mice. Antibodies derived from genotype C HBV prevented genotype A infection. HB0478 antibody prevented infection by a vaccine escape mutant. When the titer of anti-HBs is high, it is thought that anti-HBs derived from any genotype HBV can prevent HBV infection. Abbreviations: HBV, hepatitis B virus; anti-HBs, hepatitis B surface antibody; PCR, polymerase chain reaction; HB, hepatitis B.

**Figure 2 vaccines-08-00456-f002:**
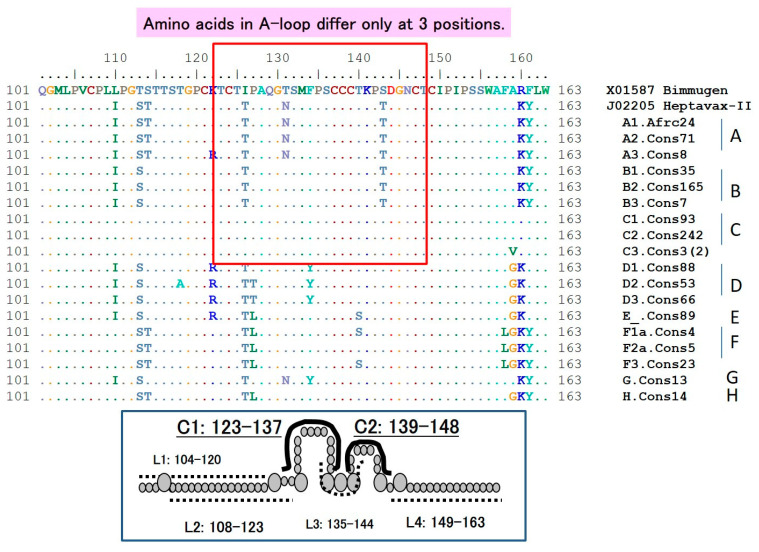
Sequence of the S-region “a” determinant. Neutralizing (protective) antibodies induced by vaccination target the conformational epitope of the “a” determinant, which is located in the S-region and retains many amino acids including Cys but with some mutations. The second hydrophilic loop (139 to 147 or 149 aa) is the major target for neutralizing anti-HBs produced following natural infection or vaccination. Abbreviations: Cys, cysteine; aa, amino acid; anti-HBs, hepatitis B surface antibody.

**Table 2 vaccines-08-00456-t002:** Major studies on vaccine escape mutant infection in children.

Publication	Country/Region	Reference	VEM/HBV Infection (%)	Status of HBV Infection	Immunization	Site of Amino Acid Substitution
1984 *	Taiwan	Hsu et al. [122]	8/103 (8%)	Persistence	Before vaccine	T126A (1), M133L (1), F134L (1), C138S (1), T140R (1), T140I (1), T143M (1), D144A (1)
1989 *	Taiwan	Hsu et al. [122]	10/51 (20%)	Persistence	HBIG+vaccine	T126A (2), P127T (1), Q129H (1), S143W (2), G145R (3), W156L (1)
1994 *	Taiwan	Hsu et al. [122]	9/32 (28%)	Persistence	HBIG+vaccine	T125A (1), P120Q + P127T (1), T126A + T143M (1), T126S + D144H (1), D144H + G145R (1), T140P (1), N146S (1), T148I (1), C147R + C149R (1)
1999 *	Taiwan	Hsu et al. [123]	3/13 (23%)	Persistence	HBIG+vaccine	T131I (1), G145R (2)
2004 *	Taiwan	Hsu et al. [124]	7/31 (23%)	Persistence	HBIG+vaccine	T126A (2), M133T (1), F134L + T148A (1), G145R (1), G145A (1), W156C (1)
1997	Taiwan	Hsu et al. [125]	1/7 (14%)	Acute or fulminant hepatitis	Vaccine	T126A + G145R (1)
1997	Taiwan	Hsu et al. [125]	5/15 (33%)	Persistence	HBIG+vaccine	T126A (2), Q129R (1), G145R (2)
2012	Taiwan	Chen et al. [126]	8/25 (32%)	Persistence	HBIG+vaccine	K122R (1), I126T (1), G145R (6)
2013	Taiwan	Wen WH et al. [127]	3/10 (30%)	Persistence	HBIG+vaccine	Q129H + T140S (1), P142L + G145R (1), G145R (1)
1992	Japan	Fujii et al. [128]	1/2 (50%)	Persistence	HBIG+vaccine	G145R (1)
1995	Japan	Hino et al. [129]	2/2 (100%)	Persistence	HBIG+vaccine	G145R (2)
1996	Japan	Miyake et al. [130]	8/46 (17%)	Persistence	HBIG+vaccine	I/T126S (3), T140S (3), G145R (1), G145K (1)
1997	Japan	Matsumoto et al. [131]	1/2 (50%)	Acute hepatitis	HBIG+vaccine	P120Q + G145R (1)
2016	Japan	Komatsu et al. [132]	5/25 (20%)	Persistence	HBIG+vaccine	I/T126S (1), G130N (1), G145R (2), G145K (1)
1995	Singapore	Oon et al. [133]	16/41 (39%)	Persistence	HBIG+vaccine	I/T126A (1), Q129H (1), M133L (1), D144A (1), G145R (10), G145R + D144A (1), G145R + P142S (1)
1997	England, Wales	Ngui et al. [134]	2/17 (12%)	Persistence	HBIG+vaccine	P120Q + Y134F + D144A (1), I126N (1)
1998	China	He et al. [135]	4/24 (17%)	Persistence	HBIG+vaccine	I/T126S (1), Q129H (1), Q129L (1), G145R (1)
2005 *	China	Yan et al. [136]	8/131 (6%)	Persistence	HBIG+vaccine	I126S (1), I126S + T131N + M133T (1), T131P (1), M133T (1), G145A (1)
2006 *	China	Yan et al. [136]	10/101 (10%)	Persistence	HBIG+vaccine	I126S (2), P127T (2), T131P (3)
2007 *	China	Yan et al. [136]	11/113 (10%)	Persistence	HBIG+vaccine	I126S (2), I126S + T131N + M133T (1), G145A (1)
2008 *	China	Yan et al. [136]	9/136 (7%)	Persistence	HBIG+vaccine	I126S (1), D144E (1), G145A (3)
2009 *	China	Yan et al. [136]	19/206 (9%)	Persistence	HBIG+vaccine	I126S (3), I126N (1), P127T (2), Q129H (2), M133I + D144A (1), D144A (1), D144N + G145R (1), G145A (1)
2010 *	China	Yan et al. [136]	7/75 (9%)	Persistence	HBIG+vaccine	I126S (1), I126N + P127T(1), D144E (1), G145R (1)
2011 *	China	Yan et al. [136]	13/102 (13%)	Persistence	HBIG+vaccine	T126A (1), I126S (1), P127T (2), Q129H (1), D144A (1), G145A (1),
2012 *	China	Yan et al. [136]	8/78 (10%)	Persistence	HBIG+vaccine	T131N (1), F134L (1), G145R (1), G145A (1)
2013 *	China	Yan et al. [136]	12/135 (9%)	Persistence	HBIG+vaccine	I126S + G130E (1), G145A (6),
2004	Pacific islands	Basuni et al. [137]	0/22 (0%)	Persistence	Vaccine	-
2016	Indonesia	Purwono et al. [138]	6/61 (12%)	Persistence	Vaccine	P120S + A159V (1), M133L (1), M133T + C147S (1), T140I (2), S155F (1)

* Surveillance; Abbreviations: VEM, vaccine escape mutant; HBIG, hepatitis B immunoglobulin.

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
