# Peer review of "Cross-Protection of Hepatitis B Vaccination among Different Genotypes"

_vaccines, 2020, doi:10.3390/vaccines8030456_

Round 1
Reviewer 1 Report
July 28, 2020
Review for Vaccines.
Manuscript entitled “Cross-protection of hepatitis B vaccination among different genotypes.
Comment
This is an excellent review on the general concern of whether or not the various vaccines prepared from the different HBV genotypes can protect all genotypes. They showed that the cross-genotype protection is possible if anti-HBV level after vaccination is high. The importance of high anti-HBV level is re-emphasized. Therefore, the measurement of antibody titer after the completion of vaccination is very important.
Their concern of the need for a booster 20 years after the initial vaccination, not only for special population but also for general population is well presented and many of healthcare workers would agree with them in most instances.
With the authors’ detailed and itemized review, this thorough review is highly educational for all healthcare workers especially for those engaged in the management of HBV related medical field.
- Cross-protection of hepatitis B vaccination among different genotypes is possible if antibody production level is high regardless of the genotype. There have been concerns whether or not there is a need to make the individual vaccine specific to the genotype. The authors answered and solved this concern.
- There has been concern and controversy whether or not the booster after many years (this time 20 yers) after initial vaccination is necessary. We do see absence of detectable antibody among the healthy adults who received HBV vaccination during childhood and had documented antibody production after vaccination.
There has been a question of whether or not a booster. would be beneficial for this population.
In this review, the authors supported our opinion of the need for booster for the general population.
Author Response
Responses to the Reviewer 1s’ comments
This is an excellent review on the general concern of whether or not the various vaccines prepared from the different HBV genotypes can protect all genotypes. They showed that the cross-genotype protection is possible if anti-HBV level after vaccination is high. The importance of high anti-HBV level is re-emphasized. Therefore, the measurement of antibody titer after the completion of vaccination is very important.
Their concern of the need for a booster 20 years after the initial vaccination, not only for special population but also for general population is well presented and many of healthcare workers would agree with them in most instances.
With the authors’ detailed and itemized review, this thorough review is highly educational for all healthcare workers especially for those engaged in the management of HBV related medical field.
Cross-protection of hepatitis B vaccination among different genotypes is possible if antibody production level is high regardless of the genotype. There have been concerns whether or not there is a need to make the individual vaccine specific to the genotype. The authors answered and solved this concern.
There has been concern and controversy whether or not the booster after many years (this time 20 years) after initial vaccination is necessary. We do see absence of detectable antibody among the healthy adults who received HBV vaccination during childhood and had documented antibody production after vaccination.
There has been a question of whether or not a booster would be beneficial for this population.
In this review, the authors supported our opinion of the need for booster for the general population.
Response: We thank you very much for your comments. We would like to state our opinion.
The principle of our review is that when we have a sufficient titer of anti-HBs, it does not need to be necessarily derived from a genotype-specific HB vaccination.
As stated in the CDC guideline, if acquired general immunity is established in the general population, it is not necessary for all to be measured anti-HBs titer and receive booster vaccination. We think that all we have to do is to keep individual immunity in high risk groups against HBV.

Reviewer 2 Report
This article written by Takako Inoue and Yasuhito Tanaka is a well written article describing the very interesting problematic of multi-genotypes vaccination.
This article need major revision to be fully suitable for publication.
Major:
Even if this article could be considered as a good review, I am pretty surprised that I cannot see any reference to PRISMA structural recommandation.
Part 2 could be more easily understood if the table 1 is completed with important data (as reference, respective prevalence/incidence, major clinical specificity, major biological finding as recombination or escape variants). Moreover, to easily understand the differnce between serological types, and to introduce further discussion, a scheme that summarize difference between them would be usefull.
Part 2.2 is incomplete as these sentence suggest the presence of subdivision of the paragraph (that is missing)
Paragraph 3.1. could be completed to fully understand the problem of HBV transmission. Give respective transmission rates for sexual practices, nosocomial transmission depending of the route,...)
Part 3.2 and 4.1.: It could be interesting to discuss the difference between vaccinal recommandation of endemic and non-endemic countries
AS this parameters are critical for the cross-protection, the authors have to give more details about the "prominent factors for improving the seroprotection rate" described from line 161 to 164.
Part 5.1. would be more easily read if contain a schematic representation of the binding sites for example. The cross protection could be explained by a (quasi-) similarity for example.
Minor:
Line 46 : give more details about the genetic diversity (percentage, hotspot location in viral genome...)
line 104 : give (numerical) precision about the "high concentration of antiHBs
Line 143 : please consider the website as a reference to include in the respective part of the manuscript.
Line 143 : Give details about the reason that justify the worlwide use of the HB vaccines containing HBsAg generated from genotype A2.
First paragraph of the part 5 contain repetition. please simplify.
Line 185-186 : this sentence is misleading. Are we reading an original article or a review?
Line 227-228 : please give details about the low/high anti-HBs titers? is the limit defined as 10?
Line 272-273 : simplify the sentence indicating only "G587A mutation" and "G145R substitution". Scientist not need more details to understand.
Line 292-293 : this point is clearly a good limit of the referenced article. Nevertheless, and to understand your purpose, it could be interesting to think with absolute values.
Table 2 is truncated and so could not be fully read. Modify.
Author Response
Responses to the Reviewer 2s’ comments
We thank the Reviewer 2 for his/her insightful comments, which have helped improve our paper.
This article written by Takako Inoue and Yasuhito Tanaka is a well written article describing the very interesting problematic of multi-genotypes vaccination.
This article need major revision to be fully suitable for publication.
Major:
Even if this article could be considered as a good review, I am pretty surprised that I cannot see any reference to PRISMA structural recommandation (sic).
Response: We thank you for your suggestion. According to your comments, we have referred to reports on the effectiveness of HB vaccination in China (lines 191-197).
Part 2 could be more easily understood if the table 1 is completed with important data (as reference, respective prevalence/incidence, major clinical specificity, major biological finding as recombination or escape variants). Moreover, to easily understand the differnce (sic) between serological types, and to introduce further discussion, a scheme that summarize difference between them would be usefull (sic).
Response: According to your suggestion, Table 1 was changed. It contains not only geographical distribution but references, major clinical features, and major biological findings (lines 93-95). Information of serological types was also added to Table 1 (lines 93-95). In addition, we added the subsection “Major biological features related to differences in genotype” (lines 96-120)
Part 2.2 is incomplete as these sentence suggest the presence of subdivision of the paragraph (that is missing)
Response: This part should have been deleted from the original template received from MDPI. This is our mistake. Now, this part has been deleted.
Paragraph 3.1. could be completed to fully understand the problem of HBV transmission. Give respective transmission rates for sexual practices, nosocomial transmission depending of the route,...)
Response: According to your advice, we added the sentences described below.
“The primary risk factor associated with HBV sexual transmission is unprotected sex with a partner with HBV infection (heterosexual or homosexual). In the situation, 26.1% of sexual partners had evidence of past and/or current HBV infection (Tufon et al. 2019). Meanwhile, the risk of HBV transmission is estimated at 19%-30%, if a non-immuned individuals against HBV are exposed to the blood from the patient who is HBeAg-positive or shows an HBV DNA > 106 IU/mL (Coppola et al. 2016)” (lines 126-130).
Part 3.2 and 4.1.: It could be interesting to discuss the difference between vaccinal (sic) recommandation of endemic and non-endemic countries
Response: As we described in the first manuscript, in 1992, the World Health Organization (WHO) recommended that all countries should introduce universal HB vaccination into their routine immunization programs (lines 153-155). Our opinion is that every individual in the world should receive an HB vaccine.
AS this parameters are critical for the cross-protection, the authors have to give more details about the "prominent factors for improving the seroprotection rate" described from line 161 to 164.
Response: As we described in the first manuscript, prominent factors for improving the seroprotection rate are an additional dose, an additional three-dose series, an increased vaccine dose, changing the route of administration, new adjuvants, and granulocyte-macrophage colony stimulating factor (GM-CSF). According to your comment, we added the most common strategy for low- and non-responders (lines 213-219).
Part 5.1. would be more easily read if contain a schematic representation of the binding sites for example. The cross protection could be explained by a (quasi-) similarity for example.
Response: In the first manuscript, we presented sequences of the S-region "a" determinant of various HBV genotypes (Figure 2). A schema of the A-loop has also been shown. We have described this as follows:
“… in the region of A-loop, the second hydrophilic loop (139 to 147 or 149 amino acid) is the major target for neutralizing anti-HBs produced following natural infection or vaccination” (lines 308-310).
Minor:
Line 46 : give more details about the genetic diversity (percentage, hotspot location in viral genome...)
Response: According to your advice, we added information regarding genetic diversity including the locations at which mutations often occur (lines 49-53, Table 1 [lines 93-95]).
line 104 : give (numerical) precision about the "high concentration of antiHBs (sic)
Response: According to your comment, with reference to the attached document of hepatitis B immunoglobulin (HBIG), we added the specific amount of anti-HBs that are each time (lines 145-146).
Line 143 : please consider the website as a reference to include in the respective part of the manuscript.
Response: We moved the website information to the reference list (line 182).
Line 143 : Give details about the reason that justify the worlwide (sic) use of the HB vaccines containing HBsAg generated from genotype A2.
Response: According to your comment, we modified the sentence as described below.
“Because available data show that current HBV-A2 vaccines are highly effective in preventing infections and clinical disease caused by all known HBV genotypes, recombinant HB vaccines containing HBsAg generated from HBV genotype A2 have been used worldwide” (lines 182-185).
First paragraph of the part 5 contain repetition. please simplify.
Response: According to your suggestion, we simplified the first paragraph of 5.1 (lines 244-252).
Line 185-186 : this sentence is misleading. Are we reading an original article or a review?
Response: This study was performed by us. The subtitle has been changed so that the readers will not misunderstand it (lines 242-243).
Line 227-228 : please give details about the low/high anti-HBs titers? is the limit defined as 10?
Response: Based on a previous report (Kato et al, 2015), we added the definitions of low/high anti-HBs titers as follows: low titer, < 30 mIU/mL and high titer, ≥ 50 mIU/mL (line 269).
Line 272-273 : simplify the sentence indicating only "G587A mutation" and "G145R substitution". Scientist not need more details to understand.
Response: According to your advice, we simplified the sentence (lines 330-331).
Line 292-293 : this point is clearly a good limit of the referenced article. Nevertheless, and to understand your purpose, it could be interesting to think with absolute values.
Response: According to your comment, we added the absolute values for the prevalence of mutants (lines 351-352).
Table 2 is truncated and so could not be fully read. Modify.
Response: According to your advice, we modified Table 2 (lines 396-399).
We look forward to hearing from you regarding our submission. We would be happy to respond to any further questions and comments you might have.

Round 2
Reviewer 2 Report
Dear Authors, thanks again for your very complete revision of your manuscript.
I think that including these new information, the latter is suitable for publication in the present form.